# Scientific Literature Database Coverage of Randomized Clinical Trials for Central Serous Chorioretinopathy

**DOI:** 10.3390/jpm13060983

**Published:** 2023-06-12

**Authors:** Lars C. Boberg-Ans, Oliver N. Klefter, Marie L. R. Rasmussen, Elon H. C. van Dijk, Yousif Subhi

**Affiliations:** 1Department of Ophthalmology, Rigshospitalet, DK-2600 Copenhagen, Denmark; lars.christian.boberg-ans@regionh.dk (L.C.B.-A.); oliver.niels.klefter.01@regionh.dk (O.N.K.); marie.louise.roed.rasmussen@regionh.dk (M.L.R.R.); 2Faculty of Health and Medical Sciences, University of Copenhagen, 2200 Copenhagen, Denmark; 3Department of Ophthalmology, Leiden University Medical Centre, 2333 ZA Leiden, The Netherlands; e.h.c.van_dijk@lumc.nl; 4Department of Ophthalmology, Alrijne Hospital, 2353 GA Leiderdorp, The Netherlands; 5Department of Clinical Research, University of Southern Denmark, 5230 Odense, Denmark

**Keywords:** central serous chorioretinopathy, randomized clinical trials, indexing, biomedical literature database, evidence synthesis methodology

## Abstract

Background: Systematic literature searches are the cornerstone of systematic reviews. In this study, we evaluated database coverage of randomized clinical trials for central serous chorioretinopathy (CSC). Methods: We searched 12 databases (BIOSIS Previews, CINAHL, the Cochrane Central, Current Contents Connect, Data Citation Index, Derwent Innovations Index, EMBASE, KCI-Korean Journal Database, MEDLINE, PubMed, SciELO Citation Index, and Web of Science Core Collection) on 10 April 2023 for randomized clinical trials for CSC. After identifying all eligible studies across all databases, we investigated the coverage of these studies within each database, including the coverage of any combination of two databases. Results: The 12 databases yielded 848 records for screening, of which 76 were randomized clinical trials for CSC. No single database provided full coverage. The most comprehensive coverage was provided by EMBASE (88%), the Cochrane Central (87%), and PubMed (75%). A combined search in the Cochrane Central and PubMed led to complete coverage (100%) while reducing the number of records for screening from 848 to 279. Conclusions: Systematic review search design should include multiple databases. For randomized clinical trials for CSC, the combination of the Cochrane Central and PubMed provides an excellent balance between coverage and workload.

## 1. Introduction

A comprehensive scientific literature search is the cornerstone of systematic reviews, for clinical guideline development and for an important amount of clinical academic work. A general recommendation is that more than one scientific literature database should be employed for systemic searches. The Cochrane Handbook states that for all Cochrane Reviews, the Cochrane Central and MEDLINE should be searched as a minimum, and preferably also EMBASE [1]. These three databases are three out of many more accessible databases. If the ambition is to provide a comprehensive review, an update of current knowledge, and a foundation upon which guidelines can be formed, at least theoretically, the literature searches should include even more databases.

However, there are also drawbacks to conducting searches in an increasing number of databases. Unlike PubMed, which is freely accessible, other databases may require access, which constitutes an economical and practical barrier. Another important aspect is that employing more databases increases the amount of work needed to screen for irrelevant records. Conducting a systematic review requires a substantial amount of work, of which screening of records constitutes an important proportion of that work. In our personal experiences, screening of records from 10+ databases can take days and even weeks [2,3,4]. This work is typically performed by clinicians outside of their routine hours, pro bono, which further stresses the need to optimize the workload and avoid unnecessary tasks. If the coverage of individual scientific literature databases was known, both alone and in combination with each other, it could potentially allow for a more effective study design of systematic reviews, at least for the parts dealing with literature search and screening of records. 

Central serous chorioretinopathy (CSC) is the fourth most prevalent exudative maculopathy (after neovascular age-related macular degeneration, diabetic macular edema, and retinal venous occlusion) [5,6]. CSC is predominantly seen in men and in individuals aged 30–60 years, and corticosteroid exposure is the most important risk factor [5,6,7,8]. Symptoms of CSC include metamorphopsia, blurred central vision, hypermetropic refractive change, and changes in color perception [6]. Optical coherence tomography allows examination of the macular anatomy and reveals the subretinal fluid, serous pigment epithelium detachments, and features that are characteristic of the pachychoroid disease spectrum [9,10,11]. Although the acute subtype of CSC can be observed as it can resolve spontaneously in many cases [6], chronic CSC needs treatment to avoid photoreceptor damage, irreversible vision loss, and reduced quality of life [6,12,13]. CSC is a topic in which many treatment modalities have been explored [14]. Although photodynamic therapy using verteporfin provides effective resolution of subretinal fluid and improvement of visual acuity [14], good long-term outcomes [15], and a desirable safety profile [16,17], there is currently a global shortage of verteporfin which limits the therapeutic use of photodynamic therapy for CSC [18]. Therefore, CSC is a disease for which there is currently a strong interest in various available treatment modalities and, therefore, in conducting systematic reviews of these treatment modalities. Further, CSC is a disease that can wax and wane, which stresses the need to focus on randomized clinical trials with an appropriately designed control or placebo group to allow evaluation of treatment efficacy [14].

In this study, we explored the scientific literature database coverage of randomized clinical trials of CSC. We first conducted a comprehensive search using 12 scientific literature databases, and then evaluated the coverage of individual databases alone and in combination to understand the most effective approach to identify all available randomized clinical trials for CSC. 

## 2. Materials and Methods

### 2.1. Study Design

This was a cross-sectional study of the literature across multiple scientific databases of medical literature. All study data are based on the indexing of published literature. According to Danish law, such studies without any sampling of patient data do not require institutional review board approval.

### 2.2. Eligibility Criteria

Eligible records were defined as those randomized to any intervention for treatment of CSC. Intervention was defined as any effort made with the intention of changing clinical outcomes of patients with CSC. We did not restrict the definition of CSC, subtype of CSC, or outcome of interest in relation to CSC. We only considered randomized clinical trials. Studies without original data, protocol publications, conference abstracts, and publications not in English were not considered eligible. We did not restrict date of publication, journal of publication, or author location.

### 2.3. Search in Literature Databases 

On 10 April 2023, we searched a total of 12 literature databases using the phrases “central serous chorioretinopathy” combined with “randomized”/“randomised”, using Boolean operators as appropriate. The details of the literature databases are provided below:BIOSIS Previews: BIOSIS Previews indexes data beginning from year 1926 to present, covering life science and biomedical science literature [19]. This database indexes journals, meetings, books, and patents, and includes clinical and experimental research, as well as methods, instrumentation, and animal studies [19].CINAHL: The Cumulative Index to Nursing and Allied Health Literature (CINAHL) indexes literature from journals on nursing, allied health, biomedicine, and healthcare [20]. Coverage for some journals dates back to the year 1937 [20].Cochrane Central: The Cochrane Central Register of Controlled Trials indexes records from multiple databases on randomized and quasi-randomized clinical trials in clinical medicine [21]. The database regularly searches multiple databases, including the trial registration database ClinicalTrials.gov to extract and index relevant clinical studies [21].Current Contents Connect: The Current Contents Connect indexes records for publications primarily in a range of disciplines, including the fields of biology and medicine [22]. Data covered start from the year 1998 to the present [22].Data Citation Index: Data Citation Index indexes records linked to data in journals, books, and conference proceedings. Data covered start from year 1900 to the present [23].Derwent Innovations Index: Derwent Innovations Index covers millions of records related to patent-related data from the year 1963 to the present [24].EMBASE: Excerpta Medica Database (EMBASE) is a biomedical and pharmacological database that tracks records in journals from year 1947 to present [25]. The database indexes include MEDLINE but also extend to a range of biomedical journals not covered by MEDLINE [25].KCI-Korean Journal Database: The KCI Korean Journal Database contains bibliographic information for scholarly literature published in the Republic of Korea, and the database is managed by the National Research Foundation of Korea [26]. Data covered start from the year 1980 to present [26].MEDLINE: Medical Literature Analysis and Retrieval System Online (MEDLINE) is a database developed and maintained by the United States National Library of Medicine [27]. The MEDLINE database had several predecessors, of which the oldest was the Index Medicus, which was established in 1870 [27]. The MEDLINE database provides systemic coverage of biomedical journals from year 1965 to present, but also older indexing for select journals with records dating back to year 1781 [27].PubMed: The PubMed database is developed and maintained by the United States National Center for Biotechnology Information and indexes a wide range of biomedical literature [28]. The database indexes include MEDLINE but also extend to a range of biomedical journals not covered by MEDLINE [28].SciELO Citation Index: The SciELO Citation Index is a multidisciplinary database with indexing of research in journals from Latin America, Spain, Portugal, the Caribbean, and South Africa [29]. Data covered start from year 2002 to present [29].Web of Science Core Collection: The Web of Science Core Collection contains records from several journal citation indexes, which include journals in a wide range of disciplines [30]. Coverage for some indexes dates to the year 1900 [30].

Documentation of searches in individual databases is provided as Appendix A.

### 2.4. Data Preparation 

Records were extracted from individual databases and subsequently imported to EndNote X9.3.1. for Mac (Clarivate Analytics, Philadelphia, PA, USA). We then removed duplicates, irrelevant records, and records deemed not eligible. Remaining records were deemed randomized clinical trials related to CSC and considered eligible for the next step of the study. For all eligible records identified, we went back to all individual databases and investigated whether the individual record was indexed in the individual databases.

### 2.5. Data Analysis and Statistics 

For each database record list, we investigated the coverage as the percentage of eligible studies that were present in the extracted database record list. We made a 2 × 2 contingency table for each database to investigate number of relevant records included in the database extract (true positives), irrelevant records in the database extract (false positives), relevant records not included in the database extracts (false negatives), and irrelevant records not in the database extract (true negatives). We then investigated sensitivity (true positives/(true positives + false negatives)), specificity (true negatives/(true negatives + false positives)), and accuracy (sensitivity × prevalence + specificity × (1 − prevalence)). We then investigated the efficacy of combining any two literature databases.

## 3. Results

### 3.1. Coverage of Individual Databases

The 12 databases yielded a total of 848 records for screening (Table 1). 

Of the 848 records, 547 records were duplicates, 225 records were not randomized clinical trials for CSC, and 76 records were randomized clinical trials. References of individual randomized clinical trials and their findings are summarized in Appendix A. 

We then explored the coverage of the 76 records identified as randomized clinical trials for CSC in each database. No single database included all of the randomized clinical trials. The most comprehensive coverage was provided by EMBASE (n = 67, 88%), the Cochrane Central (n = 66, 87%), and PubMed (n = 57, 75%). No studies were identified from Data Citation Index, Derwent Innovations Index, and SciELO Citation Index. A detailed summary of the coverage rates in individual databases is provided in Table 2.

### 3.2. Accuracy of Individual Databases 

For individual databases, we defined relevant records as true positives, irrelevant records as false positives, relevant records not in the individual database yield as false negatives, and irrelevant records not in the individual database yield but indexed in other databases as true negatives. Based on these definitions, we summarized the distribution of true positives, false positives, false negatives, and true negatives for each database (Table 3).

These numbers were included for the calculation of the sensitivity, specificity, and accuracy of the individual databases (Table 4). 

Databases with more than 50% sensitivity were EMBASE (88%), Cochrane Central (87%), PubMed (75%), Web of Science Core Collection (67%), and Current Contents Connect (58%). Among these databases with >50% sensitivity, specificity from highest to lowest was seen in Current Contents Connect (93%), Web of Science Core Collection (91%), Cochrane Central (90%), PubMed (90%), and EMBASE (85%). The accuracy reflects both the sensitivity and specificity and should be interpreted cautiously, as it can be high in databases with very low sensitivity. However, when focusing on databases with >50% sensitivity, the ranking order of accuracy was Cochrane Central (90%), Current Contents Connect (90%), Web of Science Core Collection (89%), PubMed (88%), and EMBASE (85%).

### 3.3. Efficacy of Combining Two Databases 

The efficacy of combining two databases for sensitivity, specificity, and accuracy was explored in Table 5. Maximum sensitivity was obtained for the combination of Cochrane Central × PubMed (100%), which also indicates that this combination led to a complete coverage of all randomized clinical trials. Meanwhile, this combination led to a relatively high specificity (82%) and accuracy (83%), which indicates an overall small number of irrelevant studies in the search record.

## 4. Discussion

In this study, we report that no single database contains more than 88% of the randomized clinical trials for CSC. This is insufficient for a comprehensive systematic review. If meta-analyses are based on such insufficient samples of available randomized clinical trials, there is a risk of providing results that may not represent the full picture of what is known. If clinical guidelines are based on such insufficient samples of available randomized clinical trials, there is a risk of missing important aspects of treatment that needs to be considered. A key finding of this study is that we confirm the notion expressed in the Cochrane Handbook [1], which is that literature searches should be performed in multiple databases, and that this is also the case for CSC. 

We also demonstrate that a combined search in the Cochrane Central and PubMed includes all available randomized clinical trials for CSC. Although it cannot necessarily be guaranteed that future randomized clinical trials for CSC will be included in this approach, it shows that a complete or near-complete coverage can be obtained with certain combinations that allow for a more effective search workload. 

Although this is the first study of its kind in ophthalmology and retinal sciences, our findings in this study are in line with previous research in other areas of clinical medicine. In psychiatric research, McDonald et al. demonstrated that no single database provided sufficient coverage of the field, and that combining PsycLIT (later merged into PsycINFO) with EMBASE led to a coverage of 91% of the journals in the field [31]. In the field of obesity prevention policy research, Hanneke and Young reported that PubMed indexed 76% of the papers in the field [32], which is a similar coverage rate when compared to the field of CSC in our study. For systematic reviews of interventions in hypertension, Rathbone et al. reported the highest study coverage in the databases EMBASE (69%), the Cochrane Central (60%), Database of Abstracts of Reviews of Effects (a now discontinued database which is indexed in the Cochrane Central) (57%), MEDLINE (57%), and PubMed (53%); and suggested that multiple databases should be searched when systematically searching for systematic reviews [33]. Royle and Waugh showed that for randomized clinical trials that investigate cost-effectiveness, searching a single database did not yield sufficient coverage, but searching more databases in addition to the combination of the Cochrane Central, MEDLINE, EMBASE, Web of Science Core Collection, and BIOSIS Previews had limited value [34]. Levay et al. evaluated the practice used by the National Institute for Health and Care Excellence (NICE) [35]. This study found that, on average, the Cochrane Library, EMBASE, and MEDLINE contributed 76.8% of the eligible studies, while other databases and other study identification techniques contributed 11% and 12.2%, respectively [35]. 

The strengths and limitations of this study must be acknowledged when interpreting the results. First, one important limitation of this study is that this study can only provide a snapshot of current coverage and cannot predict any changes in indexing patterns of scientific literature databases or any changes in publishing patterns for CSC. Further, it should be noted that our study focuses on randomized clinical trials and did not investigate the large amount of retrospective clinical studies of CSC. Indexing patterns of these studies may not necessarily follow those of randomized clinical trials. Justesen et al. reported that the CINAHL had full coverage of qualitative studies in the field of diabetes mellitus [36]. Kelley and Kelley reported that PubMed and the Cochrane Central had the highest coverage rates for randomized clinical trials in the field of arthritis [37]. These two studies elegantly demonstrate that certain databases may have better coverage of journal types with higher publication rates of specific types of studies [36,37]. Second, another important limitation is that we here consider studies published in the English language. Other patterns of database coverage are very likely for other languages. Specifically, we would expect that the KCI Korean Journal Database would identify relevant RCTs only reported in Korean, and that the SciELO Citation Index would identify relevant RCTs only reported in Spanish or in Portuguese. Third, considering the findings of previous studies, the circumstances for literature coverage for CSC may not be unique, but may represent a more general phenomenon that should be considered for any clinical systematic review of randomized clinical trials. However, this study only investigated the circumstances for CSC, and any extrapolation to other diseases should be made with care and caution.

In conclusion, we here investigated the scientific literature database coverage of randomized clinical trials for CSC. We find that for CSC, a systematic review search design should include multiple databases and, as a minimum, should include a combination of the Cochrane Central and PubMed. However, if the combination of the Cochrane Central and PubMed is present, there should be a sufficiently high degree of coverage to allow for a comprehensive search while also limiting the workload needed for record screening.

## Figures and Tables

**Table 1 jpm-13-00983-t001:** Number of records yielded from the search in individual scientific literature databases.

Database	Number of Records
BIOSIS Previews	80
CINAHL	26
Cochrane Central	141
Current Contents Connect	97
Data Citation Index	0
Derwent Innovations Index	0
EMBASE	184
KCI-Korean Journal Database	4
MEDLINE	102
PubMed	138
SciELO Citation Index	2
Web of Science Core Collection	126
Total records	848

**Table 2 jpm-13-00983-t002:** Coverage of individual databases of the 76 randomized clinical trials for central serous chorioretinopathy (CSC).

Database	Coverage of Randomized Clinical Trials of CSC
BIOSIS Previews	32 (42%)
CINAHL	12 (16%)
Cochrane Central	66 (87%)
Current Contents Connect	44 (58%)
Data Citation Index	0 (0%)
Derwent Innovations Index	0 (0%)
EMBASE	67 (88%)
KCI-Korean Journal Database	4 (5%)
MEDLINE	28 (37%)
PubMed	57 (75%)
SciELO Citation Index	0 (0%)
Web of Science Core Collection	51 (67%)

**Table 3 jpm-13-00983-t003:** Distribution of relevant, irrelevant, and identified records in individual databases for identification of randomized clinical trials for central serous chorioretinopathy (CSC).

Database	Relevant Records (TP)	Irrelevant Records (FP)	Unidentified Relevant Records (FN)	Unidentified Irrelevant Records (TN)
BIOSIS Previews	32	48	44	768
CINAHL	12	14	64	822
Cochrane Central	66	75	10	707
Current Contents Connect	44	53	32	751
Data Citation Index	0	0	76	848
Derwent Innovations Index	0	0	76	848
EMBASE	67	117	9	664
KCI-Korean Journal Database	4	0	72	844
MEDLINE	28	74	48	746
PubMed	57	81	19	710
SciELO Citation Index	0	2	76	846
Web of Science Core Collection	51	75	25	722

Abbreviations: FP = false positives; FN = false negatives; TP = true positives; and TN = true negatives.

**Table 4 jpm-13-00983-t004:** The sensitivity, specificity, and accuracy of individual databases for the identification of randomized clinical trials in central serous chorioretinopathy (CSC).

Database	Sensitivity (95% CI)	Specificity (95% CI)	Accuracy (95% CI)
BIOSIS Previews	42% (31–54%)	94% (92–96%)	90% (88–92%)
CINAHL	16% (8–26%)	98% (97–99%)	92% (89–93%)
Cochrane Central	87% (77–94%)	90% (88–92%)	90% (88–92%)
Current Contents Connect	58% (46–69%)	93% (91–95%)	90% (88–92%)
Data Citation Index	0% (0–5%)	100% (100–100%)	92% (90–93%)
Derwent Innovations Index	0% (0–5%)	100% (100–100%)	92% (90–93%)
EMBASE	88% (79–94%)	85% (82–87%)	85% (83–88%)
KCI-Korean Journal Database	5% (1–13%)	100% (100–100%)	92% (90–94%)
MEDLINE	37% (26–49%)	91% (89–93%)	86% (84–89%)
PubMed	75% (64–84%)	90% (87–92%)	88% (86–91%)
SciELO Citation Index	0% (0–5%)	100% (99–100%)	92% (90–94%)
Web of Science Core Collection	67% (55–77%)	91% (88–93%)	89% (86–91%)

Abbreviations: 95% CI = 95% confidence interval.

**Table 5 jpm-13-00983-t005:** Efficacy of combining two databases for the identification of randomized clinical trials for central serous chorioretinopathy (CSC). Numbers are stated as sensitivity in the first line, specificity in the second line, and accuracy in the third line.

Database	BIOSIS	CINAHL	CENTRAL	CURRENT	DCI	DII	EMBASE	KCI-KJD	MEDLINE	PubMed	SciELO	WOS
BIOSIS		Se 53% Sp 91% Ac 88%	Se 91% Sp 80% Ac 81%	Se 67% Sp 84% Ac 82%	Se 42% Sp 94% Ac 89%	Se 42% Sp 94% Ac 89%	Se 91% Sp 75% Ac 76%	Se 46% Sp 94% Ac 89%	Se 58% Sp 82% Ac 80%	Se 83% Sp 80% Ac 80%	Se 42% Sp 94% Ac 89%	Se 74% Sp 81% Ac 80%
CINAHL			Se 89% Sp 80% Ac 81%	Se 62% Sp 83% Ac 81%	Se 16% Sp 91% Ac 84%	Se 16% Sp 91% Ac 84%	Se 88% Sp 74% Ac 76%	Se 21% Sp 91% Ac 85%	Se 43% Sp 91% Ac 85%	Se 43% Sp 81% Ac 77%	Se 43% Sp 81% Ac 77%	Se 80% Sp 80% Ac 80%
CENTRAL				Se 93% Sp 86% Ac 87%	Se 87% Sp 98% Ac 97%	Se 87% Sp 98% Ac 97%	Se 96% Sp 75% Ac 77%	Se 87% Sp 98% Ac 97%	Se 89% Sp 85% Ac 86%	Se 100% Sp 82% Ac 83%	Se 87% Sp 98% Ac 97%	Se 95% Sp 83% Ac 84%
CURRENT					Se 58% Sp 95% Ac 92%	Se 58% Sp 95% Ac 92%	Se 89% Sp 75% Ac 76%	Se 62% Sp 95% Ac 92%	Se 62% Sp 83% Ac 81%	Se 82% Sp 80% Ac 80%	Se 58% Sp 95% Ac 92%	Se 72% Sp 80% Ac 80%
DCI						Se 0% Sp 90% Ac 82%	Se 88% Sp 74% Ac 76%	Se 5% Sp 90% Ac 82%	Se 37% Sp 80% Ac 76%	Se 80% Sp 80% Ac 80%	Se 0% Sp 89% Ac 81%	Se 70% Sp 80% Ac 79%
DII							Se 88% Sp 74% Ac 76%	Se 5% Sp 90% Ac 82%	Se 37% Sp 80% Ac 76%	Se 80% Sp 80% Ac 80%	Se 0% Sp 89% Ac 81%	Se 70% Sp 80% Ac 79%
EMBASE								Se 89% Sp 98% Ac 97%	Se 88% Sp 85% Ac 85%	Se 93% Sp 81% Ac 82%	Se 88% Sp 98% Ac 97%	Se 91% Sp 82% Ac 83%
KCI-KJD									Se 41% Sp 80% Ac 77%	Se 83% Sp 80% Ac 80%	Se 5% Sp 90% Ac 82%	Se 74% Sp 81% Ac 80%
MEDLINE										Se 80% Sp 80% Ac 80%	Se 37% Sp 93% Ac 88%	Se 71% Sp 80% Ac 79%
PubMed											Se 80% Sp 97% Ac 96%	Se 84% Sp 82% Ac 82%
SciELO												Se 70% Sp 80% Ac 79%
WOS												

Abbreviations: 95% CI = 95% confidence interval; Ac = accuracy; BIOSIS = BIOSIS Previews; CENTRAL = Cochrane Central; CURRENT = Current Contents Connect; DCI = Data Citation Index; DII = Derwent Innovations Index; KCI-KJD = Korean Journal Database; SciELO = SciELO Citation Index; Se = sensitivity; Sp = specificity; and WOS = Web of Science Core Collection.

## Data Availability

Publicly available datasets were analyzed in this study. Data sources are stated in the Section 2.

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
