# Peer review of "Scientific Literature Database Coverage of Randomized Clinical Trials for Central Serous Chorioretinopathy"

_jpm, 2023, doi:10.3390/jpm13060983_

Round 1
Reviewer 1 Report
This is a timely review with excellent guidance for those researchers who might want to embark on a comprehensive literature review for their work.
Manuscript is written clearly and is relevant and well structured.
There are no figures. The tables appropriately convey the data and are easy to interpret and understand. Conclusion that the Cochran and PubMed databases provide highest yield for this subject area is valuable information for researchers.
Author Response
Reviewer #1 general comments:
This is a timely review with excellent guidance for those researchers who might want to embark on a comprehensive literature review for their work.
Manuscript is written clearly and is relevant and well structured.
There are no figures. The tables appropriately convey the data and are easy to interpret and understand. Conclusion that the Cochran and PubMed databases provide highest yield for this subject area is valuable information for researchers.
Authors’ response:
We thank the reviewer for the positive feedback and for the valuable comments.
Reviewer 2 Report
The authors must be congratulated for conducting this nice study. A key finding of this study is that literature searches should be performed in multiple databases and that this is also the case for CSC. However, some of the points where improvements can be done are
1. The 76 studies should be listed in a supplementary file.
2. THe Clinical findings of the major and important studies should be mentioned and discussed.
3. The current Discussion is a very limited one. Please make it more elaborate. The important clinical findings of the studies can be discussed.
4. The limitations of the study should be mentioned.
Author Response
Reviewer #2 general comments:
The authors must be congratulated for conducting this nice study. A key finding of this study is that literature searches should be performed in multiple databases and that this is also the case for CSC. However, some of the points where improvements can be done are:
Authors’ response:
We thank the reviewer for the positive feedback and for the valuable comments.
Reviewer #2 comment #1:
1. The 76 studies should be listed in a supplementary file.
Authors’ response:
These studies are now listed in supplementary file 2.
Reviewer #2 comment #2:
2. THe Clinical findings of the major and important studies should be mentioned and discussed.
Authors’ response:
Thank you for this comment. We have not mentioned nor discussed the individual studies since the topic of our study is the database coverage rather than the individual findings of the included studies. To accommodate best possible way, we now summarize the findings of the studies in the supplementary file 2.
Reviewer #2 comment #3:
3. The current Discussion is a very limited one. Please make it more elaborate. The important clinical findings of the studies can be discussed.
Authors’ response:
Thank you for this comment. We have not mentioned nor discussed the individual studies since the topic of our study is the database coverage rather than the individual findings of the included studie. To accommodate best possible way, we now summarize the findings of the studies in the supplementary file 2.
Reviewer #2 comment #4:
4. The limitations of the study should be mentioned.
Authors’ response:
We now discuss the limitations of the study more extensively.